| Open Peer Review | Clinical Microbiology | Methods and Protocols

# Advancing COVID-19 diagnostics: rapid detection of intact SARS-CoV-2 using viability RT-PCR assay

Judith M. J. Veugen,[1,2,3,4] Tom Schoenmakers,[5,6] Inge H. M. van Loo,[3,4] Bart L. Haagmans,[7] Mathie P. G. Leers,[5,8] Mart M. Lamers,[7] Mayk Lucchesi,[3,4] Bas C. T. van Bussel,[4,9,10] Walther N. K. A. van Mook,[9,11] Rudy M. M. A. Nuijts,[1,2,12] Paul H. M. Savelkoul,[3] Mor M. Dickman,[1,2] Petra F. G. Wolffs[3,4]

**ABSTRACT**  Severe acute respiratory syndrome coronavirus 2 (SARS-CoV-2) causes coronavirus disease 2019 (COVID-19). Commonly used methods for both clinical diagnosis of SARS-CoV-2 infection and management of infected patients involve the detection of viral RNA, but the presence of infectious virus particles is unknown. Viability PCR (v-PCR) uses a photoreactive dye to bind non-infectious RNA, ideally resulting in the detection of RNA only from intact virions. This study aimed to develop and validate a rapid v-PCR assay for distinguishing intact and compromised SARS-CoV-2. Propidium monoazide (PMAxx) was used as a photoreactive dye. Mixtures with decreasing percentages of intact SARS-CoV-2 (from 100% to 0%) were prepared from SARS-CoV-2 virus stock and a clinical sample. Each sample was divided into a PMAxx-treated part and a non-PMAxx-treated part. Reverse transcription-PCR (RT-PCR) using an in-house developed SARS-CoV-2 viability assay was then applied to both sample sets. The difference in intact SARS-CoV-2 was determined by subtracting the cycle threshold ($Ct$) value of the PMAxx-treated sample from the non-PMAxx-treated sample. Mixtures with decreasing concentrations of intact SARS-CoV-2 showed increasingly lower delta $Ct$ values as the percentage of intact SARS-CoV-2 decreased, as expected. This relationship was observed in both high and low viral load samples prepared from cultured SARS-CoV-2 virus stock, as well as for a clinical sample prepared directly from a SARS-CoV-2 positive nasopharyngeal swab. In this study, a rapid v-PCR assay has been validated that can distinguish intact from compromised SARS-CoV-2. The presence of intact virus particles, as determined by v-PCR, may indicate SARS-CoV-2 infectiousness.

**IMPORTANCE**  This study developed a novel method that can help determine whether someone who has been diagnosed with coronavirus disease 2019 (COVID-19) is still capable of spreading the virus to others. Current tests only detect the presence of severe acute respiratory syndrome coronavirus 2 (SARS-CoV-2) RNA, but cannot tell whether the particles are still intact and can thus infect cells. The researchers used a dye that selectively blocks the detection of damaged virions and free RNA. They showed that this viability PCR reliably distinguishes intact SARS-CoV-2 capable of infecting from damaged SARS-CoV-2 or free RNA in both cultured virus samples and a clinical sample. Being able to quickly assess contagiousness has important implications for contact tracing and safely ending isolation precautions. This viability PCR technique provides a simple way to obtain valuable information, beyond just positive or negative test results, about the actual risk someone poses of transmitting SARS-CoV-2 through the air or surfaces they come into contact with.

**KEYWORDS**  SARS-CoV-2, viability PCR, PMAxx, viral shedding, intact virus particles, infectiousness

Address correspondence to Petra F. G. Wolffs, p.wolffs@mumc.nl.

The authors declare no conflict of interest.

See the funding table on p. 8.

Severe acute respiratory syndrome coronavirus 2 (SARS-CoV-2) is an enveloped RNA virus that causes coronavirus disease 2019 (COVID-19), which has rapidly become a global health problem since its first identification in Wuhan, China (1). On 11 March 2020, COVID-19 was declared a pandemic by the World Health Organization (WHO).

The clinical presentation of SARS-CoV-2 infection ranges from asymptomatic to severe COVID-19 (1). Human-to-human transmission of SARS-CoV-2 occurs primarily through respiratory droplets, but SARS-CoV-2 has also been detected in various other body fluids from COVID-19 patients, including blood, stool, saliva, and conjunctival samples (2, 3).

The period of viral shedding plays a critical role in the transmission of diseases and determines an individual's infectious period, which is crucial for the successful implementation of disease control strategies (4). Importantly, viral shedding dynamics are influenced by viral factors along with host factors such as age, sex, and immune status (5–7).

The most commonly used methods to detect SARS-CoV-2 RNA include reverse transcription-polymerase chain reaction (RT-PCR) (8, 9) and reverse transcription-loop mediated isothermal amplification (RT-LAMP) (10, 11). Molecular techniques are fast, sensitive, and specific, compared to culture-based methods, but have a limitation. They cannot determine whether samples that tested positive for SARS-CoV-2 contain intact infectious virus particles or viral RNA remnants only. Interpretation of infectiousness based on a positive PCR test may therefore be inaccurate (12). In a systematic review and meta-analysis performed by Cevik et al., comprising 43 studies and 3,229 individuals, viral RNA was detected by nucleic acid amplification tests (NAATs) in the upper respiratory tract for a mean duration of 17 days. In contrast, it was not possible to culture viable virus after day 9 of illness despite continued high viral loads indicated by low cycle threshold ($Ct$) values (13). These results suggest that prolonged PCR detection of SARS-CoV-2 might be due to the detection of viral RNA remnants.

Currently, culture-based methods are still the gold standard to assess the viability of SARS-CoV-2 (14, 15). However, due to the high pathogenicity and transmissibility of SARS-CoV-2 these techniques require at least a biosafety level 3 (BSL-3) laboratory, are time-consuming, and have impaired sensitivity compared to molecular techniques. As a result, wide-scale is hampered in clinical diagnostic settings. Consequently, there is a need for a rapid and feasible technique that can differentiate between intact and compromised SARS-CoV-2.

Viability PCR (v-PCR) is an easily implementable technique in which samples are pre-treated with a photoreactive dye such as propidium monoazide (PMA) that specifically binds to freely accessible nucleic acid (NA) or NA of compromised viral and bacterial cells, thereby preventing NA amplification (16–20). PMA is photoactivated upon irradiation with blue light to obtain a covalent bond between PMA and NA, but PMA cannot penetrate intact cell membranes. The NA bound to PMA is not amplifiable in the PCR, while NA of intact viral or bacterial cells is unaffected by PMA and can be amplified with PCR. In addition, several studies found that adding low concentrations of surfactants such as Triton X-100 and SDS could enhance penetration of PMA into compromised virus particles (17, 19, 20). Nevertheless, the use of this technique has not been studied in depth for enveloped viruses and its effectiveness could not be confirmed for enveloped RNA avian influenza virus (AIV) and enveloped DNA infectious laryngotracheitis virus (ILTV) (21, 22). The initial findings of this technique for SARS-CoV-2 are nevertheless promising, indicating its potential effectiveness (20). The aim of this study was to develop and validate a sensitive v-PCR assay, which is rapid and easy to perform, that can distinguish between intact and compromised SARS-CoV-2.

## MATERIALS AND METHODS

### Samples

Calu-3 cells were maintained in Opti-MEM I (1×; Gibco, Thermo Fisher Scientific, Waltham, MA) + Gibco GlutaMAX (Gibco; Thermo Fisher Scientific, Waltham, MA) supplemented with 10% fetal bovine serum (FBS; heat-inactivated for 30 min at 56℃; Sigma-Aldrich Corporation, Saint Louis, MO), penicillin (100 IU/mL), and streptomycin (100 IU/mL). Cells were kept at 37℃ in a humidified CO2 incubator. Viruses were grown to passage 3 on Calu-3, harvested 48–72 h post-infection, cleared for 5 min at 1,000 × $g$, aliquoted, and stored at −80℃ until use. All work with infectious SARS-CoV-2 was performed in a Class II Biosafety Cabinet under BSL-3 conditions at Erasmus Medical Center, Rotterdam, the Netherlands.

SARS-CoV-2 RNA was obtained from a positive residual nasopharyngeal swab from a patient with a high viral load, who was included in the CoLaIC study. Ethical approval has been obtained from the medical ethics committee (Medisch Ethische Toetsingscommissie 2020–1565/3 00 523) of the Maastricht University Medical Centre+ (Maastricht UMC+), which will be performed based on the Declaration of Helsinki. During the pandemic, the board of directors of Maastricht UMC + adopted a policy to inform patients and ask their consent to use the collected data and to store samples for COVID-19 research purposes. To increase the volume available for analysis and facilitate the experimental procedures, the clinical sample was diluted in virus transport medium (VTM; HiMedia Laboratories GmbH, Einhausen, Germany). The sample was assumed to comprise approximately 100% intact virus particles. The delta $Ct$ value was calculated to determine the ratio of intact vs compromised virions.

### Virus inactivation and sample preparation

To prepare mixtures with different concentrations of intact virus, part of the virus stock as well as the clinical sample were inactivated by dividing the samples into two parts, an intact part and a non-intact part. About 125 µL of RNAse inhibitor (20 U/µL; Invitrogen, Thermo Fisher Scientific, Waltham, MA) was added to 5 mL of intact virus pool. The non-intact virus pool was heat-inactivated by subjecting five tubes containing 1 mL of viral suspension to 95℃ for 5 min. After this, the non-intact virus tubes were re-pooled and 125 µL of RNAse inhibitor (20 U/µL; Invitrogen, Thermo Fisher Scientific, Waltham, MA) was also added to this virus pool. The non-intact part was mixed with the intact part to prepare mixtures with concentrations of 100%, 50%, 10%, 1%, 0.1%, and 0% intact SARS-CoV-2.

### PMA treatment

Samples were pre-treated with the photoreactive dye PMAxx (Biotium, Inc., Hayward, CA) to prevent amplification from free accessible RNA or RNA from compromised cells. Accordingly, each mixture containing a different concentration of intact SARS-CoV-2 was divided into a tube to which PMAxx was added (+PMA) and a tube without PMAxx (−PMA); 200 µL sample was added to both tubes and additionally 0.005% SDS and 100 µM PMAxx were added to the +PMA tubes. The +PMA tubes were vortexed and incubated for 30 min at 37℃, while 200 µL of lysis buffer (Chemagic Viral DNA/RNA kit, PerkinElmer, Inc., Waltham, MA) was added to the −PMA tubes. After incubation, +PMA tubes were subjected to photolysis for 10 min using the PMA-Lite (Biotium, Inc., Hayward, CA) after which 200 µL of lysis buffer was added to the +PMA tubes (Fig. 1).

To evaluate the potential impact of the 30-min incubation at 37℃, 0.005% SDS, and the PMA lite, experiments were performed using extracted SARS-CoV-2 RNA. The v-PCR assay was performed for each step separately to assess its individual influence.

### Nucleic acid isolation

All samples were analyzed at the Department of Medical Microbiology, Infectious Diseases & Infection Prevention, Maastricht University Medical Center+, Maastricht, the

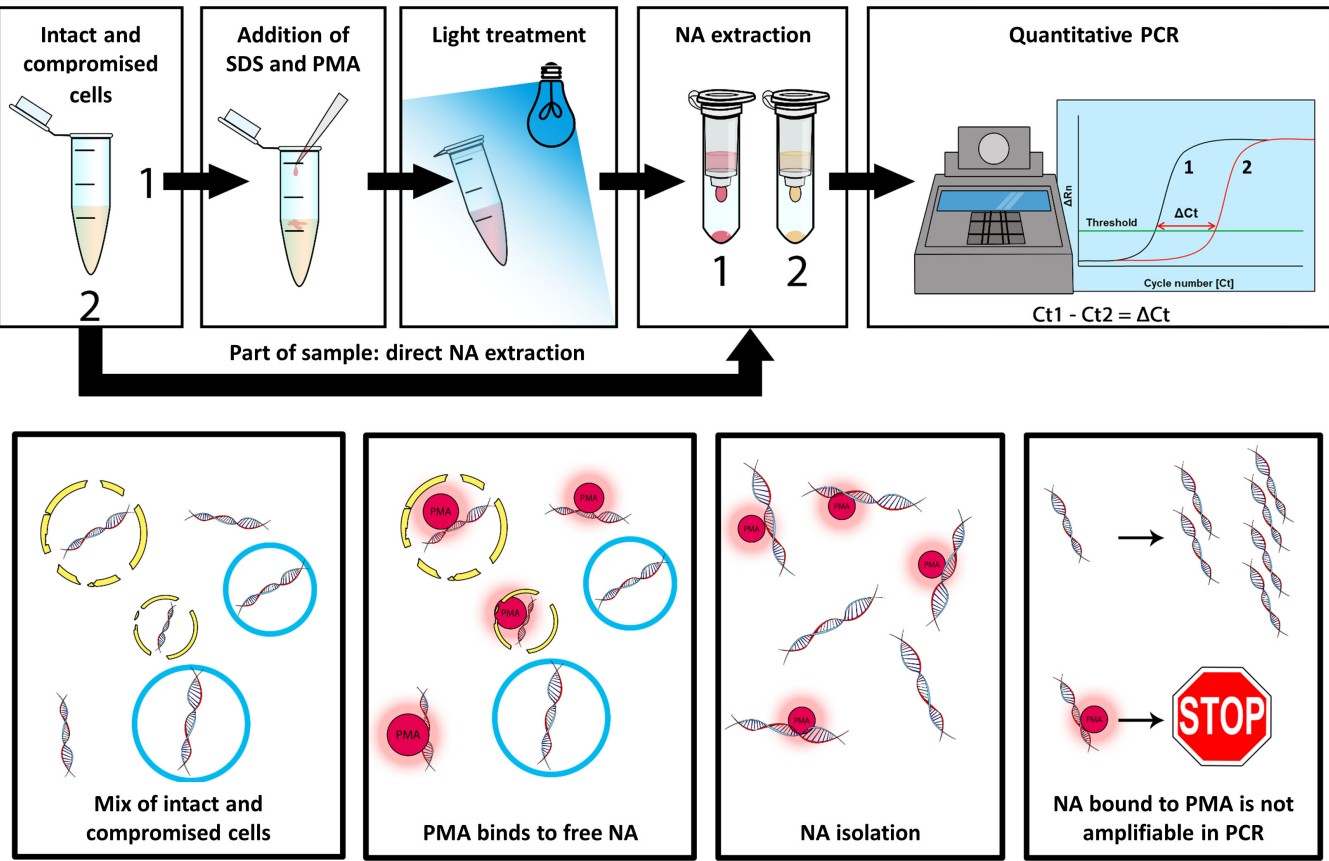

**FIG 1** Schematic representation of the viability PCR (v-PCR). Propidium monoazide (PMA) binds to free RNA. RNA bound to PMA is not amplifiable in the PCR, while RNA of intact virus is unaffected by PMA and can be amplified with PCR. The delta cycle threshold ($\Delta Ct$) value is calculated by subtracting the $Ct$ value of the PMAxx-treated sample from the $Ct$ value of the non-PMAxx-treated sample.

Netherlands. Viral RNA was extracted using the MagNA Pure 96 system (Roche Diagnostics GmbH, Mannheim, Germany). Extraction was performed using the MagNA Pure 96 DNA and Viral NA Small Volume Kit (Roche Diagnostics GmbH, Mannheim, Germany) and the Pathogen Universal 200 Protocol (MagNA Pure 96 system, Roche Diagnostics). A 200-µL sample was extracted and eluted in 50 µL elution buffer and diluted with 50 µL water for molecular biology (VWR International, Radnor, PA).

## RT-PCR analysis

RT-PCR assays were carried out on a Quantstudio 5 system (Applied Biosystems, Thermo Fisher Scientific, Waltham, MA) using an in-house developed viability assay targeting the envelope (E) gene or the nucleocapsid (N1) gene. The forward and reverse primer sequences for the v-PCR E gene were 5′-CGGAAGAGACAGGTACGTTAATAG-3′ and 5′-AGACCAGAAGATCAGGAACTCTA-3′, respectively. The probe sequence was 5′−6-FAM-ACACTAGCCATCCTTACTGCGCTTCG-BHQ-1−3′. For the v-PCR N1 gene, forward and reverse primer sequences were 5′-GACCCCAAAATCAGCGAAAT-3′ and 5′-TCTGGTAGCTCTTCGGTAGTA-3′, respectively and the probe sequence was 5′-ABY-ACCCCGCATTACGTTTGGTGGACC-BHQ-2−3′. The forward and reverse primer sequences and the probe sequence for the E gene PCR used in the standard diagnostics were 5′-ACAGGTACGTTAATAGTTAATAGCGT-3′, 5′-ATATTGCAGCAGTACGCACACA-3′ and 5′−6-FAM ACACTAGCCATCCTTACTGCGCTTCG-BHQ-1−3′, respectively. The final reaction volume was 20 µL and contained 5 µL 4× TaqPath 1-Step RT-qPCR Master Mix (Applied Biosystems, Thermo Fisher Scientific, Waltham, MA), 5 µL primer/probe mix, and 10 µL sample. Cycling conditions consisted of uracil-N-glycosylase (UNG) incubation at 25°C for 2 min, RT incubation at 50°C for 30 min,

enzyme activation at 95°C for 2 min, and 42 cycles of denaturation at 94°C for 3 s and annealing/extension at 60°C for 30 s. The $Ct$ value for each sample was determined using Quantstudio Design and Analysis Software v1.5.2.

## RESULTS

### Validation v-PCR

The first goal of this study was to demonstrate that PMAxx can block free RNA. The second step involved validating the v-PCR on cultured virus. As a final step in this validation, the viability PCR was performed on a clinical sample to determine if the v-PCR results in cultured virus reflect the results in clinical samples.

### Single-stranded RNA

To evaluate the effectiveness of the viability PCR on single-stranded SARS-CoV-2 RNA, primers and probes targeting both the E gene and the N1 gene were developed. The primers and probes are designed to detect larger PCR fragments of 210 base pairs (bp) and 263 bp, respectively, to create sufficient binding sites for PMAxx (23). The results of these v-PCRs were compared to an E gene PCR detecting a smaller PCR fragment of only 113 bp, which is used in routine clinical diagnostics.

The newly developed E gene and N1 gene PCRs resulted in delta $Ct$ values of −14.7 (log reduction of 4.5) and −14.2 (log reduction of 4.3), respectively, while the E gene PCR used in the routine diagnostics resulted in a delta $Ct$ value of only −6.6 (log reduction of 2).

Based on these results, we decided to continue with the E gene (210 bp) PCR to assess the viability of SARS-CoV-2.

### Functionality PMA

To demonstrate the functionality of PMAxx and assess the potential influence of the 30-min incubation at 37°C, 0.005% SDS, and the PMA lite, duplicated experiments were performed using extracted SARS-CoV-2 RNA. The individual steps of 30-min incubation at 37°C, 0.005% SDS, and the PMA lite did not affect the v-PCR assay, as evidenced by mean delta $Ct$ values of 0.4, 0.4, and 0.5, respectively. However, an effect was observed when these steps were combined with the pretreatment using PMAxx, resulting in a mean delta $Ct$ value of >−13.5, corresponding to a mean log reduction >4.1.

### Samples

Cultured virus was used to validate the ability of the v-PCR to distinguish between intact virus particles and NA. Mixtures containing concentrations decreasing from 100% to 0% intact SARS-CoV-2 were analyzed. These serial dilutions with different ratios of intact SARS-CoV-2 showed that the delta $Ct$ values gradually increased with a decrease in the ratio of intact virus.

In cultured virus with high viral load (VL; $Ct$ value conventional SARS-CoV-2 PCR ≈ 15), mixtures showed mean delta $Ct$ values of −2.1 (100%), −3.5 (50%), −6.2 (10%), −9.4 (1%), −12.3 (0.1%), and −14.2 (0%), respectively, corresponding to mean log reductions of 0.64, 1.06, 1.88, 2.85, 3.73, and 4.3, respectively. The serial dilution showed a linear correlation between the percentage of intact SARS-COV-2 (decreasing from 100% to 0% intact SARS-CoV-2) and the delta $Ct$ value, $R^2 = 0.995$, as determined by linear regression on log-transformed values (Fig. 2A; Table S1).

In low VL cultured virus ($Ct$ value conventional SARS-CoV-2 PCR ≈ 30), mixtures showed mean delta $Ct$ values of −2.4 (100%), −3.8 (50%), −5.8 (10%), and −8.5 (1%), respectively, which result in mean log reductions of 0.73, 1.15, 1.76, and 2.58, respectively. The delta $Ct$ values were ≥−11.6 (0.1%) and ≥−11.5 (0%), respectively, resulting in mean log reductions of ≥−3.52 and ≥−3.48, respectively. The serial dilution showed a linear correlation between the percentage of intact SARS-COV-2 (decreasing from 100%

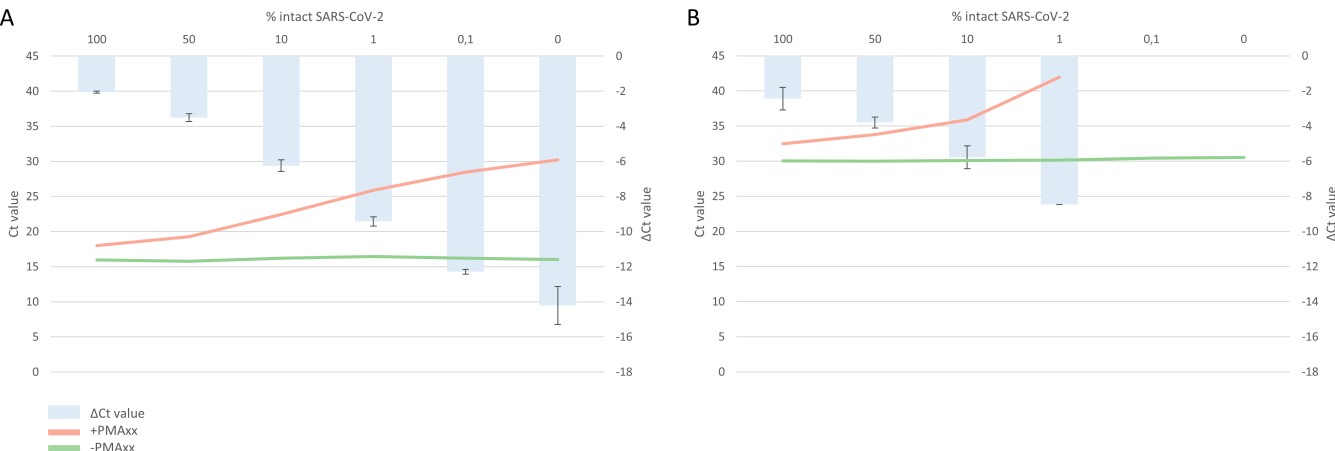

**FIG 2** Validation results of the viability PCR (v-PCR) on cultured virus with (A) high viral load (VL) and (B) low VL. Mixtures containing intact SARS-CoV-2 ranging from 100% to 0% were divided into a PMAxx-treated sample vs a non-PMAxx-treated sample. Delta cycle threshold (ΔCt) values are calculated by subtracting the Ct value of the PMAxx-treated sample from the non-PMAxx-treated sample. Ct values are shown as means; ΔCt values are shown as means ± standard deviations.

to 1% intact SARS-CoV-2) and the delta Ct value, $R^2 = 0.992$, as determined by linear regression on log-transformed values (Fig. 2B; Table S1).

To confirm whether the v-PCR is also applicable to determine intact SARS-CoV-2 in patients, the experiments were repeated in a clinical sample. Mixtures showed mean delta Ct values of −0.4 (100%), −0.8 (50%), −2.6 (10%), −4.5 (1%), −6.8 (0.1%), and −7.1 (0%), respectively, corresponding to mean log reductions of 0.12, 0.24, 0.79, 1.36, 2.06, and 2.15, respectively. The serial dilution showed a linear correlation between the percentage of intact SARS-COV-2 (decreasing from 100% to 0% intact SARS-CoV-2) and the delta Ct value, $R^2 = 0.997$, as determined by linear regression on log-transformed values (Fig. 3; Table S1).

## DISCUSSION

The purpose of this study was to validate a sensitive and rapid v-PCR assay that can differentiate between intact and compromised SARS-CoV-2. In the present study, a sensitive, rapid, and easily implementable v-PCR assay has been established. The assay effectively discriminates between intact and compromised SARS-CoV-2 based on the difference in Ct values between PMAxx-treated samples and non-PMAxx-treated samples.

The results of the present study showed a maximum mean delta Ct value of −14.2 in high VL cultured virus, indicating that PMAxx treatment of cultured inactivated high VL SARS-CoV-2 effectively blocks more than 99.99% of free RNA and RNA in compromised SARS-CoV-2. In low VL cultured virus, PMAxx treatment demonstrated the ability to block over 99.9% of free RNA and RNA in compromised virus particles. Comparatively, the delta Ct values obtained in this study were slightly higher than those reported by Hong et al., who observed a mean delta Ct value of −9.6 after PMAxx treatment of cultured inactivated SARS-CoV-2 (20). Notably, in 100% intact cultured virus, for both high and low VLs, delta Ct values of −2.4 and −2.1, respectively, were detected, indicating RNA degradation despite the assumption that the virus was fully intact. However, in the clinical sample such degradation has not been observed in 100% intact virus. Although the achieved delta Ct value was only −7.1, corresponding to a blocking efficiency of over 99% of free RNA and RNA in compromised SARS-CoV-2. Overall, the assay developed in this study is suitable for analyzing materials collected in VTM.

One limitation of this study concerns the artificial inactivation of the virus, which raises uncertainty regarding the comparability of this process to natural inactivation mechanisms. While successful inactivation was assumed, definitive proof would require culturing experiments, which were not performed. The second limitation is related to the

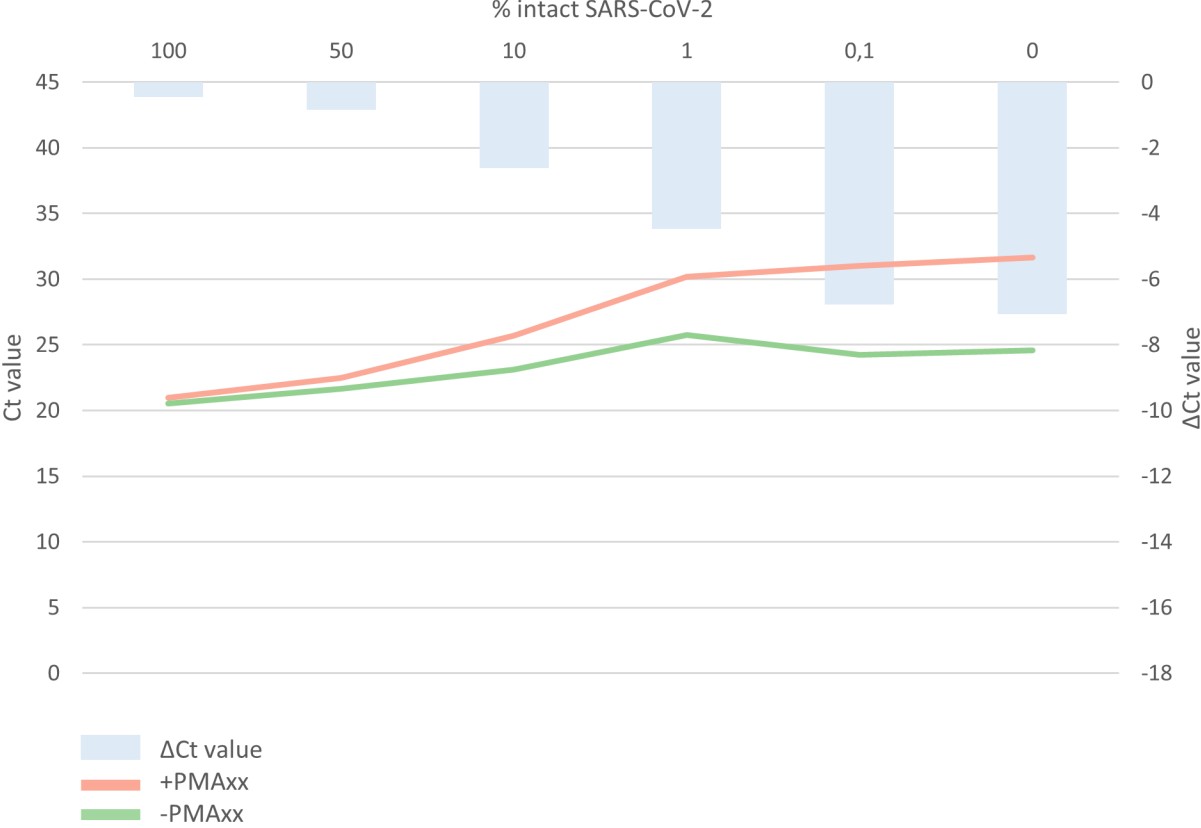

**FIG 3** Validation results of the viability PCR (v-PCR) on clinical samples. Mixtures containing intact SARS-CoV-2 ranging from 100% to 0% were divided into a PMAxx-treated sample versus a non-PMAxx-treated sample. Delta cycle threshold (ΔCt) values are calculated by subtracting the *Ct* value of the PMAxx-treated sample from the non-PMAxx-treated sample.

detection of intact virus particles. While viral integrity is a crucial factor in determining infectiousness, it is important to note that other factors, such as the presence of viral attachment proteins, also influence a virus's ability to cause infection.

A strength of this study includes the feasibility of this technique. Pretreating samples with a photoreactive dye can be a solution in situations where culture-based techniques require too much time and in conditions where culture is not possible (24). Importantly, this technique can be combined with any PCR assay while preserving the characteristics and performance of the original assay, provided the fragment is long enough for optimal pMAXX binding. Any molecular lab can pre-process samples with a photoreactive dye with a minimal impact on the current workflow. In the present study, the sample pre-processing time was limited to only 45 min, of which 40 min were incubation periods. Another strength is the validation of the v-PCR on both viral culture samples (golden standard) and patient material.

In conclusion, a sensitive and rapid v-PCR assay has been validated that can discriminate between intact and compromised SARS-CoV-2. The presence of intact virions may be a measure of SARS-CoV-2 infectiousness, providing the most optimal rapid and convenient indication of infectiousness available to date.

## ACKNOWLEDGMENTS

This publication is part of the CoLalC project (project number 10430102110002) of the COVID-19 research program which is (partly) financed by the Netherlands Organization for Health Research and Development (ZonMw). No conflicting relationship exists for any author.

## AUTHOR AFFILIATIONS

[1]University Eye Clinic Maastricht, Maastricht University Medical Center, Maastricht, the Netherlands

[2]School for Mental Health and Neuroscience (MHeNs), Maastricht University, Maastricht, the Netherlands

[3]Department of Medical Microbiology, Infectious Diseases & Infection Prevention, Maastricht University Medical Center, Maastricht, the Netherlands

[4]Care and Public Health Research Institute (CAPHRI), Maastricht University, Maastricht, the Netherlands

[5]Department of Clinical Chemistry and Hematology, Zuyderland Medical Center, Sittard-Geleen/Heerlen, the Netherlands

[6]School of Nutrition and Translational Research in Metabolism (NUTRIM), Maastricht University, Maastricht, the Netherlands

[7]Viroscience Department, Erasmus Medical Center, Rotterdam, the Netherlands

[8]Faculty of Science, Environmental Sciences, Open Universiteit, Heerlen, the Netherlands

[9]Department of Intensive Care Medicine, Maastricht University Medical Center, Maastricht, the Netherlands

[10]Cardiovascular Research Institute Maastricht (CARIM), Maastricht University, Maastricht, the Netherlands

[11]School of Health Professions Education (SHE), Maastricht University, Maastricht, the Netherlands

[12]Department of Ophthalmology, Zuyderland Medical Center, Heerlen, the Netherlands

## PRESENT ADDRESS

Mart M. Lamers, Programme in Emerging Infectious Diseases, Duke NUS Medical School, Singapore, Singapore

## AUTHOR ORCIDs

Judith M. J. Veugen http://orcid.org/0000-0002-0469-5781
Bart L. Haagmans http://orcid.org/0000-0001-6221-2015
Mart M. Lamers http://orcid.org/0000-0002-1431-4022
Petra F. G. Wolffs http://orcid.org/0000-0002-5326-3985

## FUNDING

| Funder | Grant(s) | Author(s) |
| --- | --- | --- |
| ZonMw (Netherlands Organisation for Health Research and Development) | 10430102110002 | Tom Schoenmakers |
| | | Inge H. M. van Loo |
| | | Mathie P. G. Leers |
| | | Bas C. T. van Bussel |
| | | Walther N. K. A. van Mook |
| | | Petra F. G. Wolffs |

## AUTHOR CONTRIBUTIONS

Judith M. J. Veugen, Conceptualization, Data curation, Formal analysis, Investigation, Methodology, Project administration, Resources, Validation, Visualization, Writing – original draft, Writing – review and editing | Tom Schoenmakers, Data curation, Formal analysis, Investigation, Methodology, Writing – review and editing | Inge H. M. van Loo, Conceptualization, Funding acquisition, Writing – review and editing | Bart L. Haagmans, Methodology, Validation, Writing – review and editing | Mathie P. G. Leers, Conceptualization, Funding acquisition, Writing – review and editing | Mart M. Lamers, Methodology, Validation, Writing – review and editing | Mayk Lucchesi, Investigation, Methodology, Project administration, Validation, Writing – review and editing | Bas C. T. van Bussel,

Conceptualization, Funding acquisition, Writing – review and editing | Walther N. K. A. van Mook, Conceptualization, Funding acquisition, Writing – review and editing | Rudy M. M. A. Nuijts, Investigation, Writing – review and editing | Paul H. M. Savelkoul, Conceptualization, Funding acquisition, Writing – review and editing | Mor M. Dickman, Investigation, Supervision, Writing – review and editing.

## ADDITIONAL FILES

The following material is available online.

### Supplemental Material

**Table S1 (Spectrum00160-24-s0001.docx).** Ct values of PMAxx treated and non-PMAxx-treated samples and ΔCt values.

### Open Peer Review

**PEER REVIEW HISTORY (review-history.pdf).** An accounting of the reviewer comments and feedback.

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
