## [Reviewer comments · Microbiology Spectrum]

Microbiology Spectrum

Advancing COVID-19 diagnostics: Rapid detection of intact SARS-CoV-2 using viability RT-PCR assay

Judith Veugen, Tom Schoenmakers, Inge van Loo, Bart Haagmans, Mathie Leers, Mart Lamers, Mayk Lucchesi, Bas van Bussel, Walther van Mook, Rudy Nuijts, Paul Savelkoul, Mor Dickman, and Petra Wolffs

Corresponding Author(s): Petra Wolffs, Maastricht Universitair Medisch Centrum+

Review Timeline:

Submission Date:	January 23, 2024
Editorial Decision:	February 25, 2024
Revision Received:	June 3, 2024
Accepted:	June 10, 2024

Editor: Anne Wyllie

Reviewer(s): The reviewers have opted to remain anonymous.

Transaction Report:

DOI: <https://doi.org/10.1128/spectrum.00160-24>

Re: Spectrum00160-24 (Advancing COVID-19 diagnostics: Rapid detection of intact SARS-CoV-2 using viability RT-PCR assay)

Dear Dr. Petra F.G. Wolffs:

Thank you for the privilege of reviewing your work. Below you will find my comments, instructions from the Spectrum editorial office, and the reviewer comments.

Please note comments from Reviewer 1 in particular. Additional verification of the assay with the appropriate controls should be considered.

Revision Guidelines

Sincerely,
Anne Wyllie
Editor
Microbiology Spectrum

Reviewer #1 (Comments for the Author):

To the test tubes, PMA and SDS were added and the tubes were incubated for 30 minutes at 37 degrees and subsequently subjected to photolysis, and then lysis buffer was added. The control tubes were not incubated, did not receive SDS and were not subjected to photolysis but were lysed immediately. The possible influence of the 30 minutes incubation at 37, the SDS, or

high-energy blue light, are thus not taken into account. An additional control in the form of extracted virus RNA treated with PMA and without PMA would have been appropriate to demonstrate the actual function of the PMA.

It is not clear to what extent the viral RNA becomes exposed by the heat treatment. Inactivation of the virus is not proven by plaque analysis.

Reviewer #2 (Comments for the Author):

Authors have tried a novel assays for SARS COVID . Experiment is well designed and done comprehensively, however there is a need for little more discussion on analytical sensitivity and specificity of the assay and also it's diagnostic sensitivity and specificity in comparison to the gold standard assay like Real Time PCR . This can be added as a limitation of the study also.

Response to reviewer comments

Editor

Please note comments from Reviewer 1 in particular. Additional verification of the assay with the appropriate controls should be considered.

We thank the editor for the opportunity to revise our paper. We performed an additional verification with the appropriate controls as suggested. The details of the additional verification can be found in our response to reviewer #1, point 1 section.

Reviewer #1

1. To the test tubes, PMA and SDS were added and the tubes were incubated for 30 minutes at 37 degrees and subsequently subjected to photolysis, and then lysis buffer was added. The control tubes were not incubated, did not receive SDS and were not subjected to photolysis but were lysed immediately. The possible influence of the 30 minutes incubation at 37, the SDS, or high-energy blue light, are thus not taken into account. An additional control in the form of extracted virus RNA treated with PMA and without PMA would have been appropriate to demonstrate the actual function of the PMA.

Response: We thank the reviewer for this suggestion, and we agree that additional experiments to demonstrate the actual function of PMAxx and the potential influence of the 30 min incubation at 37°C, 0.005% SDS, and the PMA lite is valuable. In response to this suggestion, we conducted additional experiments in duplicate using extracted SARS-CoV-2 RNA. The results obtained clearly indicate that the 30 min incubation at 37°C, 0.005% SDS, and the PMA lite alone do not affect the v-PCR assay. We only observed an effect when these steps were combined with the pretreatment of samples using PMAxx (Table 1).

Table 1. Cycle threshold (Ct) values and delta Ct (Δ Ct) values of SARS-COV-2 RNA to demonstrate the actual function of PMAxx. Δ Ct values are calculated by subtracting the Ct value of the treated sample from the direct lysed sample. UD, undetectable: no detectable Ct value that exceeded the threshold within 42 cycles.

Sample	Conditions				Ct value I	Δ Ct value I	Ct value II	Δ Ct value II
1	Direct lysis				23.6		23.6	
2		30 min RT			23.2	0.4	23.6	0.0
3				10 min PMA lite	22.9	0.7	23.1	0.5
4		30 min 37°C			23.2	0.4	23.2	0.4
5		30 min 37°C	0.005% SDS		23.4	0.2	23.0	0.6
6		30 min 37°C	0.005% SDS	10 min PMA lite	23.1	0.5	23.2	0.4

7	100 μ M PMAxx	30 min 37°C	0.005% SDS	10 min PMA lite	37.0	-13.4	UD	>-13.5
8	NTC (WMB)				UD		UD	

The following paragraphs were added to the manuscript:

“To evaluate the potential impact of the 30-minute incubation at 37°C, 0.005% SDS, and the PMA lite, experiments were performed using extracted SARS-CoV-2 RNA. The v-PCR assay was performed for each step separately to assess its individual influence.” (lines 138-140, materials and methods) and

“**Functionality PMAxx**

To demonstrate the functionality of PMAxx and assess the potential influence of the 30-minute incubation at 37°C, 0.005% SDS, and the PMA lite, duplicated experiments were performed using extracted SARS-CoV-2 RNA. The individual steps of 30-minute incubation at 37°C, 0.005% SDS, and the PMA lite did not affect the v-PCR assay, as evidenced by mean delta Ct values of 0.4, 0.4, and 0.5 respectively. However, an effect was observed when these steps were combined with the pretreatment using PMAxx, resulting in a mean delta Ct value of >-13.5, corresponding to a mean log reduction >4.1.” (lines 188-195, results)

2. It is not clear to what extent the viral RNA becomes exposed by the heat treatment. Inactivation of the virus is not proven by plaque analysis.

Response: We appreciate the reviewer raising this important point regarding verification of SARS-CoV-2 inactivation. As the reviewer correctly noted, we were unable to prove complete inactivation of the virus as no culturing after inactivation was performed.

During optimization of the protocol, we did attempt multiple heat inactivation protocols, including heating at 60°C for various time durations. However, we found these were insufficient as the maximum delta Ct value achieved was only -0.9. Upon heat inactivation for 5 min at 95°C, we were able to reach a maximum delta Ct value of -14.2, similar to the inactivation of naked RNA. While we assumed successful inactivation, we acknowledge we could not definitively prove this without culturing experiments and added this as a limitation to the manuscript : “While successful inactivation was assumed, definitive proof would require culturing experiments, which were not performed.” (lines 244-245, discussion)

Reviewer #2

1. Authors have tried a novel assays for SARS COVID . Experiment is well designed and done comprehensively, however there is a need for little more discussion on analytical sensitivity and specificity of the assay and also it's diagnostic sensitivity and specificity in comparison to the gold standard assay like Real Time PCR.

Response: We acknowledge the reviewer's comment regarding the need for a more comprehensive discussion on the analytical sensitivity and specificity of the v-PCR in comparison to the gold standard assay.

The sensitivity and specificity of the v-PCR are influenced by the choice of the PCR used after pMAXX treatment. The method can be combined with any PCR, provided the fragment is long enough to achieve good pMAXX binding.

Regarding the sensitivity of the current v-PCR setup, we found it to be approximately 5 copies/PCR.

To evaluate the specificity of the v-PCR, we applied the v-PCR in a large clinical study and compared the results of the v-PCR with our current routine clinical diagnostics PCR. During this comparison, no false positive results were observed with the v-PCR

We added the following text to the discussion: “Importantly, this technique can be combined with any PCR assay while preserving the characteristics and performance of the original assay, provided the fragment is long enough for optimal pMAXX binding.” (lines 251-253, discussion)

Re: Spectrum00160-24R1 (Advancing COVID-19 diagnostics: Rapid detection of intact SARS-CoV-2 using viability RT-PCR assay)

Dear Dr. Petra F.G. Wolffs:

Thank you for your thoughtful edits in response to the reviewer's comments.

Your manuscript has been accepted, and I am forwarding it to the ASM production staff for publication. Your paper will first be checked to make sure all elements meet the technical requirements. ASM staff will contact you if anything needs to be revised before copyediting and production can begin. Otherwise, you will be notified when your proofs are ready to be viewed.

Sincerely,
Anne Wyllie
Editor
Microbiology Spectrum